# Self-Antigens Targeted by Regulatory T Cells in Type 1 Diabetes

**DOI:** 10.3390/ijms23063155

**Published:** 2022-03-15

**Authors:** Angela M. Mitchell, Aaron W. Michels

**Affiliations:** Barbara Davis Center for Diabetes, University of Colorado, Aurora, CO 80045, USA; angela.mitchell@cuanschutz.edu

**Keywords:** regulatory T cells, type 1 diabetes, islet autoimmunity, antigen-specific therapy

## Abstract

While progress has been made toward understanding mechanisms that lead to the development of autoimmunity, there is less knowledge regarding protective mechanisms from developing such diseases. For example, in type 1 diabetes (T1D), the immune-mediated form of diabetes, the role of pathogenic T cells in the destruction of pancreatic islets is well characterized, but immune-mediated mechanisms that contribute to T1D protection have not been fully elucidated. One potential protective mechanism includes the suppression of immune responses by regulatory CD4 T cells (Tregs) that recognize self-peptides from islets presented by human leukocyte antigen (HLA) class II molecules. In this review, we summarize what is known about the antigenic self-peptides recognized by Tregs in the context of T1D.

## 1. Introduction

Type 1 diabetes (T1D) is an autoimmune disease that typically presents in childhood and early adulthood that results in the destruction of insulin-producing pancreatic beta cells by T cells [1,2,3,4]. The primary pathological presentation of T1D is inflammation of the pancreatic islets, termed insulitis, and it is due to the infiltration of immune cells, including CD4 and CD8 T cells along with B cells [5,6,7,8,9,10]. Therefore, there is a direct link between pathogenic T cells targeting insulin-producing pancreatic beta cells; however, regulatory T cells (Tregs) play an important role in protection from autoimmunity, including T1D. During T1D development, there is likely an imbalance between pathogenic T cells targeting pancreatic islets and Tregs that function to protect against targeting these cells.

The incidence of T1D continues to increase globally across racial and ethnic groups [11]. However, the measurement of T1D-associated antibodies in the peripheral blood allows for the prediction of disease progression through stages, as islet autoantibodies precede the development of clinical diabetes, in general, by a number of years [12,13,14]. Furthermore, a recent study has shown that short-term immunotherapy with an anti-CD3 monoclonal antibody can delay the onset of clinical disease [15], which is marked by hyperglycemia and the need for exogenous insulin treatment. Despite these clinically meaningful successes, there is a strong need in the field to specifically induce tolerance to islet antigens prior to and at disease onset. In this review, we aim to describe what is known regarding the specific antigens that are targeted by regulatory T cells in type 1 diabetes, and we will summarize ongoing antigen-specific therapeutic strategies for preventing disease onset.

## 2. T Cell Receptor–Peptide–MHC Interactions

T cells possess T cell receptors (TCRs) that recognize peptides bound to major histocompatibility complex (MHC) molecules on the surface of antigen-presenting cells (APCs) such as B cells, dendritic cells, and macrophages. TCRs consist of an alpha chain and beta chain that are generated via the recombination of noncontiguous gene segments for each chain. Both chains of the TCR include a variable (V), joining (J), and constant region, while the beta chain also contains a diversity (D) region. Therefore, the process by which TCRs are rearranged into fully functional receptors is known as V(D)J recombination. In this manner, each T cell generates a unique receptor for the recognition of a closely related set of peptides presented in the context of the MHC molecule. The interaction between TCR/peptide/MHC leads to activation of the T cells, and two major classes of T cells are involved in an adaptive immune response. Both classes recognize peptide/MHC, and cluster of differentiation 4 (CD4) T cells release cytokines to activate other immune cells and also interact with and activate B cells, while CD8 T cells act to directly kill target cells. CD4 cells can generally be subdivided into T helper type 1 (Th1) and type 2 (Th2) cells, which are distinguished by their functions and cytokine production. Th1 cells are considered proinflammatory due to their production of the cytokines interleukin-2 (IL-2) and interferon gamma (IFN-γ). Th1 cells function to direct immune responses against intracellular viral and bacterial pathogens. Th2 cells are considered more anti-inflammatory and produce the cytokines IL-4, IL-5, and IL-13 to generate immune responses directed against extracellular pathogens. Regulatory T cells (Tregs) are another subtype of CD4 T cells that suppress immune responses via multiple mechanisms, including the secretion of anti-inflammatory cytokines (e.g., IL-10 and transforming growth factor beta (TGF-β)), the expression of regulatory cell surface receptors (e.g., cytotoxic T-lymphocyte-associated protein 4 (CTLA-4)), and the direct killing of APCs via perforin and granzyme B.

T cells are directly involved in the immunopathogenesis of T1D, and the link between autoimmunity and particular MHC alleles has been well established. In fact, numerous studies have demonstrated that MHC is the major genetic determinant for T1D [16,17,18,19,20,21,22,23,24]. Conversely, the presence of diabetes-resistant MHC alleles in transgenic mouse models can both induce the deletion of autoreactive T cells and also promote the development of Tregs [19,20,21]. Therefore, the MHC genotype may determine the balance between proinflammatory and anti-inflammatory responses to a given self-antigen. In humans, MHC proteins are encoded by the human leukocyte antigen (HLA) genes, and variants of these genes can confer significant disease risk or protection from autoimmune diseases, including T1D [25]. For example, individuals with the HLA-DQ8 (DQB1*03:02) allele have an odds ratio for T1D development of 11, while those with HLA-DQ6 (DQB1*06:02) are protected from T1D with an odds ratio of only 0.03 [26,27,28]. T1D research has primarily been focused on identifying peptides that activate T cells via presentation by diabetes risk MHC class II molecules [29]. However, little is known about the mechanisms by which protective MHC molecules provide dominant protection from T1D development.

## 3. Immunologic Tolerance by Regulatory T Cells

Two types of Tregs include thymus-derived natural Tregs (nTregs) and peripherally induced Tregs (iTregs). nTregs express the transcription factor forkhead box P3 (Foxp3) and suppress other T cell subsets during inflammation, whereas iTregs are conventional T cells that are induced to express Foxp3 and become regulatory in peripheral lymphoid tissues. A subtype of iTreg cells, known as type 1 regulatory (Tr1) cells, do not express high levels of Foxp3 but rather CD49b and lymphocyte-activation gene 3 (LAG-3) and are able to produce high levels of both IL-10 and TGF-β [30,31]. Thus, Tr1 cells are potent suppressors of other T cell subsets via cytokine-mediated mechanisms. However, Tr1 cells can also mediate suppression via granzyme-mediated killing in a cell contact-dependent fashion [32]. Distinct subtypes of Tr1 cells have now been identified in both humans and mice, and the subtypes differ primarily in their cytokine secretion profiles [33].

Tregs are not only important for the return to a homeostatic state after an immune response, but they are also critical for the prevention of autoimmunity. For example, Tregs induce tolerance in the periphery by suppressing autoreactive T cells that are specific for self-tissues. However, the precise molecular mechanisms by which Tregs suppress auto-antigen-specific T cells are unknown, and it is of interest to determine which antigens are being recognized by Tregs to prevent autoimmunity. It is appreciated that T1D patients do not lack overall numbers of Tregs but likely have a functional defect in bulk Tregs, which contributes to disease development [34,35,36,37]. However, we will focus on studies that identified and evaluated islet antigen-specific Tregs in autoimmune T1D.

## 4. Antigen-Specific Tregs in Type 1 Diabetes

Although the deletion of most autoreactive T cells occurs in the thymus, self-reactive T cells do escape and are able to circulate in the periphery [38]. These escaped self-reactive T cells can be controlled by Tregs as one mechanism of peripheral tolerance. Determining the specificity of these Tregs may aid in understanding the mechanisms involved in the loss of tolerance to self-antigens that occurs during autoimmunity and identifying specific antigens and epitopes that may be utilized for antigen-specific immunotherapy to treat the underlying autoimmunity.

In T1D, Kwok et al. showed that islet-specific glucose-6-phosphatase catalytic subunit-related protein (IGRP)-specific T cells from healthy individuals and those with T1D could produce both proinflammatory (i.e., IFN-γ) and anti-inflammatory (i.e., IL-10) cytokines, indicating that antigen-specific Th1 cells and Tregs are present in the peripheral blood [39]. The IGRP-specific T cells were detected using HLA-DR4 and -DR3 tetramers, and therefore, they were of high avidity for peptide/MHC. However, because both pro- and anti-inflammatory IGRP-specific T cells were present in healthy individuals in addition to T1D patients, the results indicate that the escape of high avidity self-reactive T cells is not sufficient to cause disease. Likely, there is an imbalance in the pathogenic CD4 T cells and protective Tregs, as measured by pro- and anti-inflammatory cytokine responses, to islet self-antigens that skews the response during T1D development with a threshold needing to be met for autoreactive T cells to target pancreatic islets.

Data in support of a cytokine imbalance toward self-antigens come from studies in our laboratory that measured cytokine responses to native and mutated insulin B chain peptides using a cytokine enzyme linked immunospot (ELISPOT) assay with peripheral blood immune cells from new-onset T1D patients as well as non-diabetic controls [40]. Importantly, the majority of individuals in both groups were carrying at least one diabetes-risk HLA allele (e.g., DQ8 or DQ2). The strongest proinflammatory T cell responses were found in response to a mutated insulin peptide—much more than the native peptide sequence. The amino acid substitution in the mutated insulin B chain peptide (B22R→E) allows the peptide to bind in an otherwise unfavorable register to T1D-risk HLA molecules (i.e., DQ8 and DQ2). In addition, anti-inflammatory responses were present in the vast majority of the non-diabetic subjects and several T1D patients. Both the IFN-γ and the IL-10 responses were greater in non-diabetic individuals who carried at least one non T1D-risk DQ allele. These results indicate that in non-diabetic individuals, the presence of a diabetes protective or neutral HLA-DQ molecule may lead to a regulatory T cell response to insulin, whereas in T1D individuals, this ability may be muted or absent. Other studies using cytokine ELISPOT assays with epitopes from beta cell specific self-antigens, proinsulin, and insulinoma-associated antigen 2 (IA-2) provide similar results with a proinflammatory response in disease and regulatory response in health [41,42].

Recently, we identified both pro- and anti-inflammatory cytokine T cell responses to hybrid insulin peptides (HIPs) from longitudinal peripheral blood samples of individuals genetically at risk for T1D who either developed islet autoantibodies or remained seronegative [43]. HIPs are neoantigens that form in the lysozymes of beta cells via a covalent bond between a fragment of C-peptide and another peptide from a beta cell protein [44]. In this manner, autoreactive T cells that are specific for HIPs may escape into the periphery because these neoantigens are not presented in the thymus during T cell education. In this study, individuals who became autoantibody positive or who progressed to clinical T1D (high blood sugars requiring exogenous insulin treatment) had a predominantly proinflammatory response to the HIPs, and these responses correlated to worsening measurements of blood glucose control. Separate studies have also found T cell responses to HIPs in newly diagnosed T1D patients [45,46,47].

As islet antigen-specific Tregs are present within healthy individuals, these cells have been cloned after culture with IA-2 (IA-2_709–736_) or proinsulin peptides (B:11–30, B:9–28) [48]. Interestingly, although the cells produced IL-10 in response to the islet antigen and were able to suppress the proliferation of T cells, the study found that direct cell-to-cell contact was required for the suppression to occur. The autoantigen-specific Tregs were further able to express cytotoxic molecules (i.e., granzyme A and granzyme B) and directly kill islet autoantigen-loaded antigen-presenting cells in a perforin/granzyme-dependent manner. These results indicate that antigen-specific Tregs are potent regulators of pathogenic T cells as well as antigen presenting cells in healthy individuals.

The age of T1D onset may also help direct T cell responses to beta cell proteins, as Ueno et al. measured CD4 T cell responses to glutamic acid decarboxylase (GAD), preproinsulin, and IGRP in adult-onset versus childhood-onset T1D patients [49]. In adult patients, there was predominantly a Th1 response to IGRP, whereas those with childhood onset had a Th2 immune response with Tr1 regulatory cells. In fact, the frequency of Tr1 cells responding to IGRP in adult-onset patients was much lower than in childhood-onset patients and non-diabetic controls. These results indicate that distinct subsets of CD4 T cells may respond to IGRP differently and could influence the timing of disease onset.

Taken together, these studies indicate that during T1D disease development, there are anti-inflammatory responses to islet autoantigens; however, there is either a defect in that response or the proinflammatory immune response predominates. This concept is highlighted by data from a spontaneous animal model of autoimmune diabetes in which there are regulatory T cells within the inflamed pancreatic islets that are directed against an immunodominant peptide, insulin B:9–23, but insulitis and diabetes still develop [50]. At the current time, it is unknown and under investigation whether islet antigen-specific Tregs are present in human insulitis.

## 5. Structural Basis of a Treg T Cell Receptor Recognizing Peptide/MHC

To better understand the structural basis of a proinsulin-specific Treg, Rossjohn et al. solved the crystal structure of a Treg T cell receptor (TCR) binding to proinsulin/DR4 and compared this to the canonical TCR docking pattern for a proinsulin-specific effector T cell [51]. Proinsulin-specific Tregs were cloned and then induced to become iTregs via interaction with tolerogenic dendritic cells. Typically, the α chain of the TCR docks over the β chain of the MHC class II molecule, while the β chain of the TCR docks over the MHCα. However, this Treg crystal structure determined that the TCRα chain overlaid the MHCα chain, while the TCRβ chain overlaid the β chain of the proinsulin/DR4 complex. Thus, a possible structural mechanism exists with reversed polarity docking of the TCR that may account for the difference in response when Tregs recognize a self-antigen versus effector T cells for the same peptide/MHC complex. Although the study identified a novel docking motif for proinsulin-specific Tregs in the context of DR4, it is unknown whether this unusual docking motif extends to other Treg cell specificities at this time.

In a separate study, two murine CD8 T cell receptors specific for a nucleoprotein epitope (NP_366_) presented by the MHC class I molecule, H-2D^b^, docked with a 180 degree position relative to other CD8 TCR-peptide-MHC class I complexes [52]. Although not a Treg TCR–peptide–MHC class II interaction, the authors found that the responses were defective in downstream signal transduction and proliferative capacity. Therefore, irregular docking of a TCR on a peptide–MHC complex may play an important role in how T cells respond to various peptides, including self-peptides. There is a need in the field to understand the structural interactions of Treg TCRs with self-peptide/MHC complexes (e.g., TCR affinity and X-ray crystallography) and compare these interactions to effector T cells binding similar peptide/MHC complexes.

## 6. Diabetes-Protective MHC Molecules Present Self-Antigens to Activate Tregs

A recent study observed that both GAD- and IGRP-specific Tregs were in higher frequencies in individuals with the T1D protective haplotype of HLA-DR15/DQ6 [53]. Tetramers consisting of either DQ6 or DR15 and several different GAD or IGRP epitopes were used to stain peripheral blood from non-diabetic individuals directly ex vivo. Both effector and regulatory antigen-specific T cells were observed in individuals with the protective haplotype. Importantly, the DR15-restricted GAD- and IGRP-specific Tregs were able to suppress the proliferation of GAD- and IGRP-specific effector T cells, including those activated by the T1D-risk DQ8 molecule. These findings provide a mechanism by which dominant protection is afforded by DQ6/DR15 when the DQ8 allele is also present. Of note, the Treg suppression did not require cell-to-cell contact, but the maximal response was achieved when it was permitted, which is in agreement with separate studies of islet-antigen specific Tregs [48]. These results indicate that a higher frequency of Tregs recognizing islet autoantigens in the context of T1D-resistant MHC molecules may be responsible for protection from T1D development. A similar phenomenon is likely present in Goodpasture’s disease (autoimmune targeting of the basement membrane proteins in the lung and kidneys) in which an epitope of the α3 chain of type IV collagen (α3_135–145_) is presented by both the risk-conferring DR15 molecule and the protective DR1 molecule, albeit in different binding registers. HLA-DR15 presentation of the epitope results in a T cell response that is predominantly proinflammatory, while DR1 results in CD4+ Foxp3+ Treg responses [54]. 

Taken together, these studies indicate that antigen-specific Tregs are present in both healthy and T1D individuals. Several different peptides from beta cell antigens have been identified as epitopes for Tregs in the peripheral blood and are summarized in Table 1. However, it is unknown which of the antigen-specific Tregs are important for protection from autoimmunity, whether autoimmunity is due to defects in these Tregs, and also at what disease stage the potentially defective antigen-specific Tregs might play a role (e.g., prior to seroconversion, after seroconversion, or at clinical disease onset). Future studies are warranted to address these questions in order to understand the role of Tregs in the stages of T1D disease pathogenesis. Furthermore, identifying and characterizing antigen-specific Tregs that mediate protection from T1D and other autoimmune diseases have important therapeutic implications.

## 7. Therapeutic Strategies to Enhance Tregs in Type 1 Diabetes

Therapies to increase Treg numbers have been tested clinically in T1D, and it has been established that Tregs can selectively expand in response to the administration of low-dose IL-2 [55,56]. Several early-phase clinical trials have been completed showing safety at low doses and as a single agent that increases CD4+ CD25+ Foxp3+ cells, and larger efficacy trials are underway in newly diagnosed children and adults [57,58,59,60,61,62,63,64]. Another approach to increasing Treg numbers has focused on expanding autologous Tregs ex vivo and reinfusing these cells back into the same T1D patient from which the cells were obtained [65,66]. The goal is to expand Tregs in an antigen-independent fashion (i.e., polyclonally) so as tip the balance in favor of an anti-inflammatory environment that may lead to protection of the remaining beta cells. A more recent phase I study combined the adoptive transfer of expanded autologous polyclonal Tregs with low-dose IL-2 [67]. With the addition of IL-2, the infused Tregs and endogenous Tregs increased in numbers for approximately 6–12 weeks after the last dose of IL-2. However, activated natural killer (NK) cells, mucosal-associated invariant T cells, and populations of CD8 T cells also increased from baseline. There was no preservation of endogenous insulin production in the participants as measured by C-peptide, which is secreted 1:1 with insulin from the beta cell. These results indicate the need to more specifically target IL-2 to avoid off-target effects. As such, engineered or mutated IL-2 using IL-2 cytokine-receptor orthogonal pairs to transmit IL-2 signals can expand Tregs but not NK or CD8 T cells in murine models [68], making this an attractive therapeutic approach once translated to human IL-2. Furthermore, microRNAs, which are single-stranded non-coding RNA molecules that alter gene expression by inhibiting target messenger RNA, are also being studied as an avenue to improve Treg function and stability [69,70,71,72,73].

To improve clinical outcomes, islet antigen-specific Tregs, which are in very low frequencies in peripheral blood, may need to be selectively expanded prior to infusion. Another approach is to combine these Treg expanding therapies with agents that modulate effector T cell function, such as monoclonal antibodies targeting CD3 on T cells [15,74,75,76,77], CD2 on T cells [78,79], antithymocyte globulin [80,81], or potentially a small molecule inhibitor of effector T cells activated by the T1D-risk HLA-DQ8 molecule in T1D [82,83,84]. Ideally, the effector T cell response would be modulated or blocked first, which is followed by a Treg-inducing agent. Finally, it would be of interest to the field to combine low-dose IL-2 with an islet antigen or to expand autologous Tregs in the presence of an epitope that is capable of inducing a regulatory immune response (such as the proinsulin epitopes depicted in Figure 1).

Antigen-specific immunotherapies have been trialed to both prevent clinical T1D onset and to preserve endogenous insulin function in new-onset patients with mixed results. Preparations of insulin (intradermal, intranasal, and oral) have not been able to prevent diabetes onset in those at risk and those having islet autoantibodies [85,86,87]; however, higher doses of oral insulin may have effects in those genetically at risk for T1D prior to the development of autoantibodies [88,89]. A GAD-alum vaccine has been used in newly diagnosed patients, again with less than optimal results in terms of preserving residual beta cell function [90,91]. However, there is strong evidence to suggest that individuals having a specific T1D risk HLA haplotype, DR3/DQ2 without DR4/DQ8, benefit from therapy [92,93]. As such, a clinical trial enrolling patients with this specific HLA-DR-DQ genotype is being planned and represents a personalized medicine approach to antigen-specific immunotherapy in T1D.

Insulin- and GAD-based intervention trials have made use of whole proteins; however, others have used self-peptides as antigen-specific therapy for T1D. In newly diagnosed patients, a DR4-restricted proinsulin peptide was injected intradermally without adjuvant every two weeks over the course of several months [94]. Compared to placebo, the peptide injection stabilized C-peptide levels as well as exogenous insulin use. Furthermore, peptide-treated individuals showed an increase in both Foxp3 expression within the Treg subset and increased IL-10 production by proinsulin-stimulated CD4 T cells compared to placebo-treated individuals. Additionally, a recent phase 1 clinical trial used multiple-peptide immunotherapy with three epitopes from proinsulin and three from IA-2 that are presented by HLA-DR4, which were safe and tolerable [95]. These results indicate that peptide-specific immunomodulatory therapies may be able to induce both protective immunologic and metabolic responses even after the onset of clinical T1D.

## 8. Challenges and Gaps in the Field

Much evidence exists to suggest an imbalance of pathogenic versus Treg responses that leads to T1D. Studies evaluating effector and regulatory T cell responses to islet antigens have predominantly been performed using peripheral blood samples. Important questions remain, including whether the imbalance exists in different tissue compartments such as the pancreatic islets, exocrine pancreas, and pancreatic lymph nodes of patients. Studying tissue samples from organ donors at risk, with multiple islet autoantibodies but no overt clinical disease and with newly diagnosed T1D, will allow for important insights into these questions. Consortia such as the network for Pancreatic Organ Donors (nPOD) and the Human Pancreas Analysis Program (HPAP) are instrumental in identifying, collecting, and studying these rare tissue samples [96,97]. The advent of next-generation sequencing technologies, including those at the single-cell level (e.g., RNA sequencing, T cell receptor sequencing, and DNA methylation), provides an avenue to obtain large amounts of data on the molecular phenotype of cells within the target tissues of T1D.

Additional questions include: When in the disease process does an imbalance of effector to regulatory responses occur? Does a defect or imbalance in Tregs lead to a loss of tolerance or drive the disease toward clinical presentation after the development of islet autoimmunity, as indicated by the presence of islet autoantibodies? Longitudinal studies that follow children at-risk for T1D from birth, such as The Environmental Determinants of Diabetes in the Young (TEDDY) [98], will greatly aid in filling these gaps in the field. Answering these questions will provide important insights into the pathogenesis of T1D and likely other organ-specific autoimmune diseases.

There is the striking finding that specific MHC class II molecules, DQ6 (DQB1*06:02) and DR15, provide dominant protection from T1D development. We propose the hypothesis that DQ6 and DR15 are capable of presenting self-peptides, including those from proinsulin and other islet-associated proteins that activate Tregs. Thus, a tolerogenic environment is induced when these peptides are encountered and presented in the periphery (Figure 2), whereas T1D risk MHC class II molecules, DQ8 and DR4, present similar peptides to effector T cells with those cells trafficking to the islets and promoting the eventual destruction of pancreatic beta cells. Evaluating this hypothesis will aid in understanding the mechanisms by which islet antigen Tregs restricted to diabetes-resistant MHC class II molecules provide protection, which will help design therapies to mimic these protective mechanisms to treat the underlying autoimmunity in T1D.

## 9. Conclusions

Regulatory T cells specific to self-antigens from pancreatic insulin-producing beta-cells are present in both individuals with and without T1D. Interestingly, diabetes-resistant MHC class II molecules can also present islet antigens to Tregs, which may provide a mechanism to protect against disease development. Understanding how antigen-specific Tregs provide protection from T1D and other autoimmune diseases will aid in the development of therapies that mimic these protective mechanisms. Utilizing antigen-specific Tregs themselves or the antigens that stimulate them provide exciting avenues to induce tolerance in an organ-specific autoimmune disease such as T1D.

## Figures and Tables

**Figure 1 ijms-23-03155-f001:**
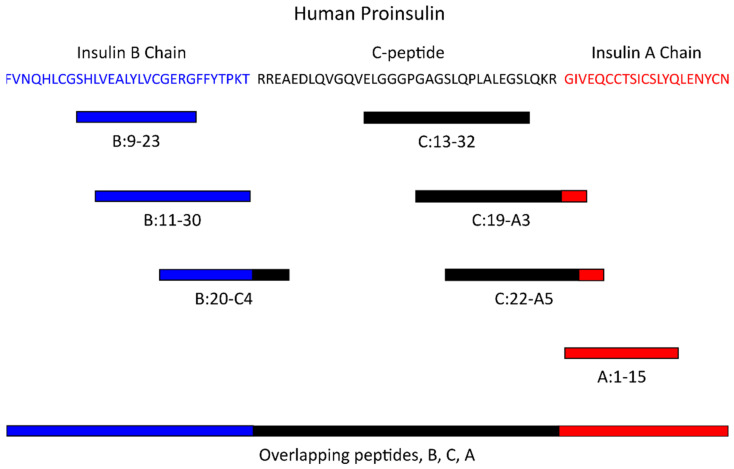
Proinsulin epitopes that induce a regulatory immune response in type 1 diabetes. Displayed is the amino acid sequence of human proinsulin and the locations of peptides recognized by regulatory T cells with accompanying citations. Blue corresponds to amino acids within the B chain, black corresponds to amino acids within the C-peptide, and red corresponds to amino acids within the A chain of proinsulin. Epitopes include B:9-23 [40,42,43,49], B:11-30 [48], B:20-C4 [49], C:13-32 [41], C:19-A3 [41,49,51,94,95], C:22-A5 [41], A:1-15 [49], and overlapping peptides from insulin B chain, C-peptide, and insulin A chain [53].

**Figure 2 ijms-23-03155-f002:**
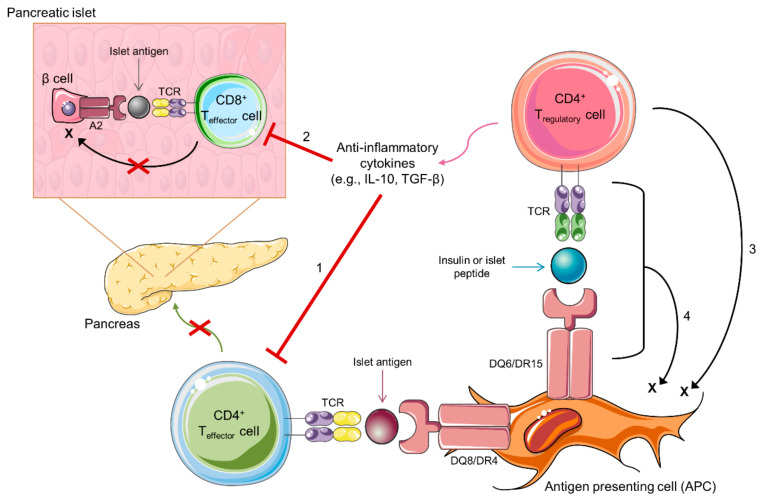
Potential mechanisms for protection from type 1 diabetes development by antigen-specific regulatory T cells. Depicted is a CD4+ regulatory T cell (Treg) recognizing an insulin or islet peptide in the context of diabetes-protective HLA-DQ6 or DR15. This interaction leads to the production of anti-inflammatory cytokines (e.g., IL-10 and TGF-β), and the activation of these antigen-specific Tregs can protect from beta cell destruction by four potential mechanisms. (**1**) CD4+ effector T cells recognize islet antigens presented by diabetes-risk conferring HLA-DQ8 or DR4 on antigen-presetting cells (APCs) and then migrate to the pancreas where they induce pancreatic beta cell destruction. Antigen-specific Tregs can suppress CD4+ effector T cells and prevent the trafficking of these cells to the pancreas via the release of anti-inflammatory cytokines. (**2**) CD8+ T effector cells in the pancreatic islets recognize islet antigens presented by HLA-A2 on pancreatic β cells, leading to the destruction of the β cells. Tregs can suppress CD8+ effector T cells and prevent β cell destruction via the release of anti-inflammatory cytokines. (**3**) Tregs can target APCs via granzyme-mediated killing in an antigen-independent manner or (**4**) potentially in an antigen-dependent manner when the cognate self-peptide is presented to the Treg T cell receptor (TCR).

**Table 1 ijms-23-03155-t001:** Select antigen-specific Treg studies in human type 1 diabetes.

Method	Autoantigen Target(s) [HLA(s)]	Primary Outcome	References
Tetramer staining	IGRP [DR3, DR4]	IL-10	[39]
Cytokine ELISPOT	Insulin B:9–23; mimotopes [DQ8]	IL-10	[40]
	HIPs [DQ8]	IL-10	[42,43]
	Proinsulin, IA-2 [DR4]	IL-10	[41]
Cloning, cytokine ELISA; flow cytometry	Proinsulin, IA-2	IL-10;	
	[DR3, DR4]	granzyme A, B;	[48]
		Treg suppression	
Cytokine beads; flow cytometry	GAD65, PPI, IGRP	IL-10; IP-10	[49]
Cloning; cytokine ELISA; flow cytometry; TCR-peptide-MHC crystallization	Proinsulin [DR4]	IL-10, granzyme B; TCR-peptide-MHCdocking	[51]
Tetramer staining, cloning, flow cytometry	GAD, IGRP, PPI, ZnT8 [DR3, DR4, DR15, DQ6]	IL-10; Treg suppression	[53]

Treg: regulatory T cell; HLA: human leukocyte antigen; ELISpot: enzyme-linked immune absorbent spot; ELISA: enzyme-linked immunosorbent assay; IGRP: islet-specific glucose-6-phosphatase catalytic subunit-related protein; HIP: hybrid insulin peptide; IA-2: islet antigen-2; GAD65: glutamic acid decarboxylase 65-kilodalton isoform, PPI: preproinsulin; ZnT8: zinc transporter 8; IL: interleukin; TCR: T cell receptor; MHC: major histocompatibility complex.

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
