# Peer review of "Self-Antigens Targeted by Regulatory T Cells in Type 1 Diabetes"

_ijms, 2022, doi:10.3390/ijms23063155_

Round 1
Reviewer 1 Report
The authors have accomplished a significant goal of summarizing the literature of known islet-derived autoantigens recognized by CD4+ Tregs presented by HLA isoforms associated with T1D promotion and resistance. The cited literature is thorough to date and accurately described/discussed. Some key concepts described in this review are: 1) Treg functions, but probably not frequencies, appear to be defective in T1D, 2) disease resistant HLA isoforms have been shown to induce Teff cell deletion and Treg expansion in an Ag-specific manner as a means to achieve tolerance, and 3) that the same islet-derived autoantigens are recognized by both pathogenic Teff and Treg cells in the same individuals with and without (healthy control) T1D, although imbalances in pro- and anti-inflammatory features are skewed between Teff and Treg cells in T1D.
The following comments should be addressed prior to acceptance for publication.
- Other than “summarizing” the literature and stating that additional research needs to be done, can the authors provide an insightful thesis/hypothesis/model of how certain islet-derived antigens presented via particular HLA isoforms might be required for Tregs to properly function and restore tolerance in T1D?
- Although "defective Tregs" (function or frequency) are associated with human T1D, is there any evidence that Treg functional defects exist as “causal to” or “a consequence of” the development of human T1D?
- One of the most significant findings supporting the hypothesis that specific self-antigen complexed with specific HLA molecules could promote skewing of Treg vs. Teff responses in T1D is by Beringer, et al. (ref. 51), via a Treg crystal structure showing that Treg TCRa chains overlay HLA (DR4)alpha chain and TCRß overlay HLAß chain, which is a reverse orientation of almost all other Teff cells. Are there any other studies that confirmed this distinction in Treg/Teff TCR recognition of HLA/Ag?
- The references after 73 required re-numbering
Reviewer 2 Report
I recommend this paper for publication only after major revisions and, unfortunately, they are really major.
Comments:
1. I suggest to emphasize the interesting recent advances published in the last 3 years. The Authors cited too many old papers (published more than 5 years ago). The literature should be updated. Much more papers from the last 3 years have to be cited and discussed to show the newest achievements in this field.
2. The numbers of references are wrong - after Ref#73 the next is 1, 2, 3, etc.
3. The major weakness of the current version is the lack of figures, schemes etc. (only one figure in the review paper is non-acceptable!). The authors should add more figures to make the ms more interesting and attractive for potential readers.
4. The authors have to discuss the issues mentioned in each section. Specific insights and actionable perspectives for the relation between self-antigens targeted by regulatory T cells and type 1 diabetes should be included.
5. Authors have to add the critical analysis of different sections in the manuscript sections. The Authors have summarized the work from different research groups, but it will be essential to discuss the advantages and limitations of the different approaches.
6. Please provide a brief summary of challenges/gaps in the field and suggest the way ahead with respect to future directions with this work.
7. Abbreviations are not appropriately defined throughout the manuscript. I recommend to add the Abbreviations section at the end of ms.
8. They should run spell-check and carefully check for typos.
Reviewer 3 Report
The authors present a nice summary on the emerging role Self-Antigens Targeted by Regulatory T Cells in Type 1 Diabetes. They shortly analysed the pathogenesis of T1D, and subsequently the protective mechanisms from developing such disease: they summarise what is known about the antigenic self-peptides recognised by Tregs in the context of T1D.
Overall, the manuscript flow properly, and the topic in interestingly. I think a few points should be addressed prior to publication.
1- In the introduction, the description of T1D is short but adequate. Please add few lines introducing the role of T cell and T reg in T1D onset.
2- Several abbreviations need to be defined or added. Please check that each population is define the first time is mentioned, Tr1, Teff..
3- Line 236: Please describe how each epitope has been designed
4- Please quickly introduce Th1 and Th2 cell populations. It can help readers
5- Section 7. Therapeutic strategies to enhance Tregs in T1D. Recently, miRNA-targeting has been identified as a strategy to improve Treg induction and stability in T1D. Please comment this approach
7- Conclusions: considering that “Regulatory T cells specific to self-antigens from pancreatic insulin-producing beta- cells are present in both individuals with and without T1D” how antigen-specific Tregs provide protection from T1D? the authors should include few hypothesis and more consideration in the Conclusions section.
8- Typo and grammar errors should be checked.
